# 'I gotta Feeling': Exploring the effects of a smartphone app (Feelee) to enhance adolescents' emotion regulation in forensic outpatient settings: A multiple single-case experimental design

Merel M. L. Leijse[1,2,3]*, Levi van Dam[2,4,5], Samantha Bouwmeester[6,7], Thimo M. van der Pol[1,2,8], René Breuk[2], Arne Popma[1,3,9]

1 Child and Adolescent Psychiatry & Psychosocial Care, Amsterdam UMC location Vrije Universiteit Amsterdam, Amsterdam, Netherlands, 2 Levvel, Academic Center for Child and Adolescent Psychiatry, Amsterdam, Netherlands, 3 Mental Health, Amsterdam Public Health, Amsterdam, Netherlands, 4 Garage2020, Dutch Innovation Network for Societal Youth Challenges, Amsterdam, Netherlands, 5 Faculty of Social and Behavioral Sciences, University of Amsterdam, Amsterdam, Netherlands, 6 Out of the Boxplot, Rotterdam, The Netherlands, 7 Department of Developmental Psychology, Tilburg School of Behavioral Sciences, Tilburg University, Tilburg, The Netherlands, 8 Forensic Mental Healthcare, Inforsa, Amsterdam, Netherlands, 9 Department of Psychiatry, Amsterdam UMC location, Vrije Universiteit, Amsterdam, The Netherlands

* m.m.l.leijse@amsterdamumc.nl

## Abstract

Adolescents in forensic outpatient care often face a complex interplay of emotional and cognitive challenges, which is also reflected in current challenges within treatment approaches. Mobile health (mHealth) interventions have shown increasing value in forensic settings, although empirical evidence remains limited. One particular mHealth app that may address current challenges in forensic outpatient care is Feelee, which provides daily emotion regulation practice through self-reported emotional check-ins and passively collected smartphone sensor data. Given the potential but still limited evidence for mHealth apps in forensic settings, this study aimed to provide first thorough evaluation of the Feelee app as an addition to treatment as usual to enhance emotion regulation skills among adolescents in forensic outpatient care. A multiple single-case experimental ABA design was applied, consisting of a 2-week baseline (phase $A_1$), 4-week intervention (phase B), and 2-week follow-up (phase $A_2$), combining quantitative and qualitative methods. Twenty-two adolescents (aged 12–23) completed daily assessments of emotion regulation. Secondary outcomes focused on emotional developmental mechanisms and treatment-related factors, measured at pre-, post-, and follow-up. Semi-structured interviews with adolescents and clinicians explored experiences with Feelee and its integration into treatment. Results showed a significant improvement in the emotional recognition during the intervention phase. No improvements were found in emotion suppression and impulse control, while reflection and distraction showed reversed outcomes. At

**Data availability statement:** All relevant data are within the manuscript and its Supporting Information files.

**Funding:** For the design and conduct of the study described in this paper, the first author received funding from Quality Forensic Care Youth ('Kwaliteit Forensische Zorg Jeugd' in Dutch). Website: https://kfzjeugd.nl/.

**Competing interests:** The authors have read the journal's policy and declare the following competing interests: LvD is an employee of Garage2020, the foundation that provides the Feelee app.

follow-up, secondary outcomes indicated increases in positive emotion differentiation, emotional awareness, and self-reflection. Treatment motivation remained stable, while therapeutic alliance improved. Qualitative findings highlighted increased emotional insight, a alongside technical difficulties and limited discussion of Feelee data during sessions. These findings suggest that Feelee may particularly be valuable in the early stages of emotion regulation by enhancing emotional recognition. Future research should explore longer-term use and actively involve clinicians in the integration of app data to maximize therapeutic relevance and impact.

## Trial registration

Central Committee on Research Involving Human Subjects NL-OMON54390 and ClinicalTrials.gov NCT06509360

## Introduction

Adolescents receiving forensic outpatient care often show difficulties in emotional and behavioral development, shaped by a complex interplay of individual and contextual risks associated with delinquent behavior [1–3]. Adolescence (age 12–24 years) is a period of substantial brain maturation, marked by heightened emotional reactivity and still-developing self-regulation systems [4,5]. At the same time, social environments become increasingly influential, as relationships with peers intensify and adolescents become more sensitive to indirect social stressors in their surroundings [6,7]. When these normative developmental challenges coincide with adverse contextual stressors, such as family instability, deviant peer influences or disengagement from school, the risk of emotional dysregulation and subsequent delinquent behavior increases [8–10]. Early and targeted interventions that address these risk factors are therefore essential to support adolescents' emotional and behavioral development and prevent escalation into adulthood.

Current interventions in forensic outpatient settings are typically built on the principles of the Risk Need Responsivity (RNR) model, which states that treatment of delinquent adolescents should correspond to the person's risk level, interventions should target criminogenic needs and align with their developmental and learning capacities [11,12]. In practice this often involves individually oriented cognitive approaches such as an offence analysis or Cognitive Behavioral Therapy (CBT), focusing on helping adolescents recognize emotional and behavioral patterns and reflect on their thoughts and responses [13]. In addition, family-based approaches such as Systemic Family Therapy (SFT) or Relational Family Therapy (RFT) aim to improve communication, strengthen interactional patterns and support more constructive relationships within the family [14,15]. While these approaches are considered evidence-based, their effectiveness remains mixed [15,16]. Meta-analyses report small to moderate effects for family-based interventions [17,18] and inconsistent outcomes for individually oriented CBT in reducing delinquent behavior [19,20].

These inconclusive results may be attributed to a potential mismatch between the demands of these interventions and the developmental and psychosocial profiles of the adolescents. Adolescence, especially among youth at risk for involvement for delinquent behavior, is a period in which vulnerability to emotional dysregulation is heightened. This can limit adolescents' emotional insight, their ability to reflect and their use of adaptive emotion regulation strategies [21–23]. Studies show that reduced sensitivity to emotional cues, such as lower amygdala reactivity and diminished physiological arousal, can make emotional signals less clear and more difficult to interpret [24,25]. In addition, a substantial body of research demonstrates that adverse contextual experiences disrupt the maturation of neural systems that support emotional learning and regulation [26–28]. Together, these vulnerabilities reduce adolescents' capacity to sufficiently benefit from current cognitively oriented interventions that depend on verbal reasoning, emotional insight, reflective processing and the capacity to understand and articulate internal experiences [29,30].

The limited insight into their own emotions and behavior not only hampers emotional learning but also reduces engagement and intrinsic motivation in treatment [31–33]. Research shows that offenders often experience a mismatch between their own goals and those of the treatment. This, together with limited choice and control, can reduce their motivation and diminishes the potential impact of the intervention [34]. These motivational difficulties are amplified when treatment is obligatory and adolescents struggle to reflect on their own role or the consequences of their actions [35,36]. Previous treatment experiences that failed to build trust and relational difficulties such as disrupted attachments may further reduce alliance quality, making sustained engagement more challenging [19,37]. As a result, forensic outpatient care is often characterized by low engagement and high rates of no-shows and dropout [31].

Against this background, mobile health (mHealth) interventions have gained increasing attention as a way to enhance traditional treatments in more personalized and continuous (24/7) interventions [38–40]. Within this development, smartphone-based mHealth apps offer accessible functionalities that enable users to easily access therapeutic content and engage in skill practice in real time and naturalistic settings [41,42]. Smartphones facilitate the collection of both active and passive data [43,44]. Active data refer to intentional self-reports such as ecological momentary assessments/ experience sampling methods (EMA/ESM), which provide daily mood reports or short questionnaires [45]. Passive data, in contrast, are automatically gathered through embedded sensors such as GPS, accelerometer or screen and can capture behavioral patterns including physical and sleep activities [44,46]. Combined, these data streams contribute to the creation of a digital phenotype, an increasingly validated method for identifying emotion and behavioral patterns [47,48]. Taken together, both the practical and data-collection functionalities of smartphone-based mHealth apps offer promising opportunities for self-monitoring and practicing emotional and behavior-related skills in daily life in ways that can support clinical care [45,49].

Despite the potential of mHealth apps for clinical purposes, research particularly in forensic care remains limited [50,51]. Review studies indicate that most available work concerns feasibility and usability rather than evaluations of clinical outcomes [39,50]. Moreover, much work is based on adult samples, overlocking the distinct developmental needs of these adolescents who remain largely underrepresented [50,52]. This gap is noteworthy since over 97% of adolescents in the Netherlands from the age of 12 use a smartphone daily [53,54]. In addition, the visual and low-language nature of mHealth apps makes them particularly suitable for transferring therapeutic skills that align with abilities and needs observed within forensic care [51,55]. Therefore, mHealth apps have the potential to better align with the emotional development and engagement challenges within forensic youth care, yet their added value remains largely unknown.

A particular mHealth app that may help to bridge this gap is the Feelee app. Feelee aims to enhance the emotion regulation abilities of adolescents by combining momentary self-reports of emotions through emoji-based check-ins with passive indicators of daily behavior such as physical activity and sleep [56]. Together, this information creates a visual overview that shows how emotions and related behaviors unfold in daily life. In contrast to many text-based mHealth apps developed for adults [50,51], Feelee uses quick emoji-based check-ins that allow adolescents to select emotions, link these emotions to situational cues to support brief reflection and gain insight into emerging patterns. This structure aligns

with emotion regulation models that describe regulation as a process of identifying emotions, interpreting their meaning and implementing responses accordingly [57–59]. The concrete and context-rich insights provided by Feelee may support adolescents' emotion regulation processes and could enhance current treatment approaches by strengthening conversations about emotions during sessions.

These insights may also strengthen treatment processes. Having access to their own emotional data has been shown to give adolescents a greater sense of autonomy and ownership, which can enhance intrinsic motivation to engage with treatment [60,61]. The visual and accessible nature of the app may also fit the developmental profile of adolescents in forensic outpatient care and offer shared, concrete material that helps clinicians and adolescents build a joint understanding of emotional patterns, which may strengthen the therapeutic alliance [62,63]. However, given the limited evidence on mHealth tools in this context, examining whether Feelee can support emotion regulation as a meaningful addition to treatment as usual represents an important next step.

## Present study

Building on earlier feasibility study [64], there is a need for a more fine-grained evaluation of whether and how mHealth apps like Feelee may support emotion regulation in the daily lives of adolescents in forensics outpatient settings and how its use may complement ongoing treatment. Therefore, the current study aimed to perform a first throughout evaluation of the Feelee app in addition to treatment as usual for adolescents receiving forensic outpatient care. A multiple single-case experimental design (SCED) with both quantitative and qualitative components was applied across two research sites. The primary objective was to examine the effects of Feelee on three core components of emotion regulation targeted by the app: emotional recognition, reflection, and management, assessed through daily measurements. As a secondary objective, we examined changes in (a) emotional developmental factors, including emotional differentiation, awareness, self-reflection and self-insight, and (b) treatment-related factors, such as treatment motivation and therapeutic alliance. These secondary constructs were assessed at pre-, post-, and follow-up assessments. The third and final objective concerned the qualitative component, which explored both adolescents' and clinicians' experiences with the use and integration of Feelee in therapeutic practice. This qualitative component aimed to provide insight into their perspectives on the perceived usefulness and clinical relevance of Feelee in supporting emotion regulation, thereby informing further refinement and implementation.

## Methods

The protocol of this study was previously published [65]. Furthermore, the methodology and results have been reported in accordance with the Single-Case Reporting Guideline in Behavioral Interventions (SCRIBE) 2016 [66].

## Study design

In this study, a non-randomized multiple Single Case Experimental Design (SCED) across multiple participants was applied (Fig 1). A SCED involves repeated measurements at the individual level, comparing phases with and without the intervention across different phases [67,68]. This design is particularly suited for examining preliminary intervention effects in small, heterogeneous clinical populations that are often difficult to include in large-scale effectiveness studies such as randomized controlled trials. In the present study, neither randomization nor concurrent baselines across participants were implemented, as the length and timing of study phases varied naturally within routine clinical practice. Blinding of participants, clinicians, and researchers was not feasible given the nature of the design. Nevertheless, SCED provides a robust framework for an in-depth, initial exploration of the potential effects of Feelee among adolescents receiving forensic outpatient care [66].

The SCED applied in the current study comprised a 2-week baseline ($A_1$), a 4-week intervention (B), and a 2-week follow-up phase ($A_2$). The phase lengths were based on SCED methodological guidelines recommending a minimum of three data points

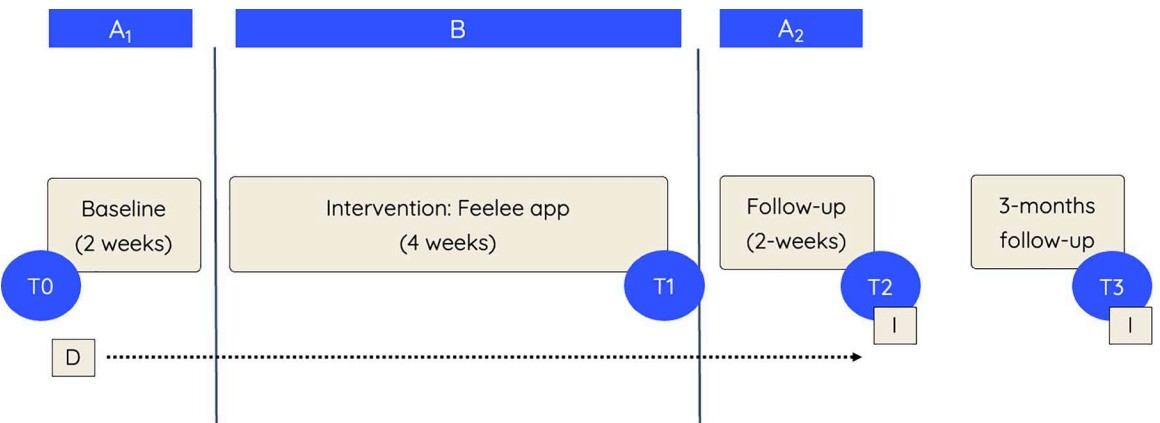

**Fig 1. Overview of Single Case Experimental Design and Measurements.** Note: A₁: baseline; B: Intervention; A₂: follow-up; T0: premeasurement; T1: postmeasurement; T2: follow-up measurement; T3: 3-months follow-up measurement; D: daily questionnaires; I: semi-structured interviews.

per phase to sufficiently capture individual changes over time, while keeping phases as short as possible to ensure feasibility and data completeness within a complex population with high potential to drop out [67,69]. Throughout all phases, participants completed a daily questionnaire assessing various aspects of emotion regulation. In addition, pre- (T0), post- (T1), and follow-up (T2) assessments were conducted to facilitate a more comprehensive analysis of the potential mechanisms underlying effects of Feelee on emotion regulation. A long-term follow-up assessment (T3) was conducted three months after the post-measurement (T2) to evaluate the potential sustained effects of Feelee. Furthermore, semi-structured interviews were administered at T2 and T3 to gain deeper insight into participants' perceived effects and experiences with Feelee as part of their treatment.

## Participants

Participants were recruited from outpatient forensic teams at two mental healthcare institutions in the Netherlands (Levvel and Inforsa) between 15 May 2023 and 15 October 2024. Since the study involved a complex population that required careful recruitment, close collaboration between researchers and care organizations was considered essential. Therefore, both sites were selected based on accessibility and existing partnerships with the research team. Furthermore, because the study aimed to gather rich, practice-based data on the effects of Feelee rather than to determine causal effectiveness, no a priori power analysis was performed.

Potential participants were initially screened for eligibility by a research staff member in consultation with the responsible care provider(s). The inclusion criteria were as follows: (1) the participant was aged between 12 and 23, (2) was receiving individual and/or family-focused counseling or treatment at Levvel or Inforsa at the time of the study, (3) had an expected treatment duration of at least three additional months, (4) owned a personal smartphone with an iOS (iPhone) or Android operating system, and (5) possessed basic proficiency in smartphone use, which was informally assessed by the clinician by confirming basic app navigation skills. Exclusion criteria included: (1) the presence of severe of serious mental disorders, such as psychosis or a high risk of suicide, (2) insufficient comprehension of spoken and written Dutch, or (3) lack of access to a personal smartphone with an iOS (iPhone) or Android operating system. In cases where eligibility was uncertain, a supervising senior clinician was consulted. Ultimately, 22 adolescents consented to participate. The overall dropout rate during the study was 36% (7 participants), which is relatively low given the forensic population [31,70]. The primary reasons for dropout included loss of interest in participating, prolonged disengagement from healthcare services (> 3 months), or discontinuation of treatment due to external circumstances. An overview of the recruitment and participation process is presented in Fig 2.

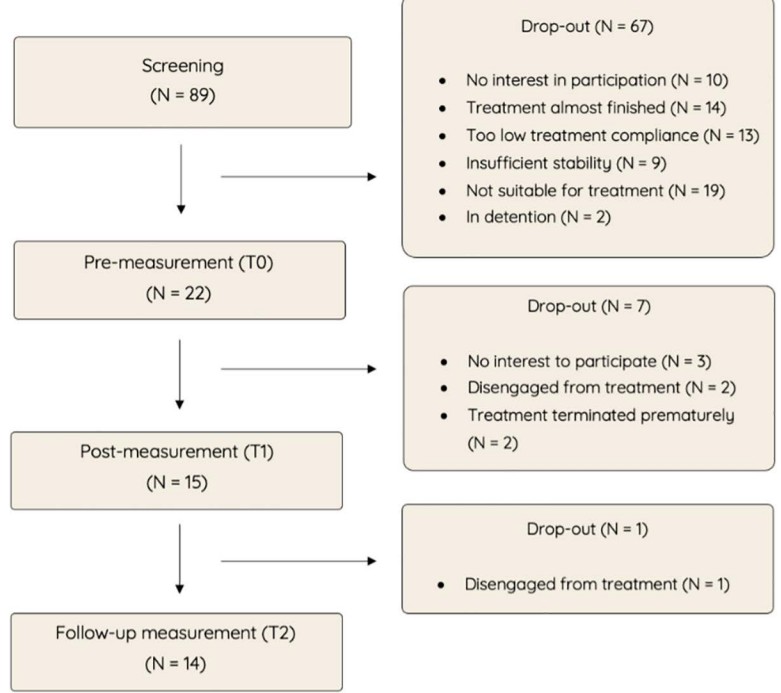

**Fig 2. Recruitment and participation flow-chart.**

## Procedure

At study onset, clinicians received a two-part training covering the study protocol, including in- and exclusion criteria, and guidance on discussion the Feelee data during the intervention phase. Clinicians conducted the initial screening and eligibility, as they maintain regular contact with the adolescents. Eligible adolescents were briefly informed about the study and the Feelee app by their clinician. Those who expressed interest were scheduled for an appointment with a researcher, typically before or after a routine treatment session, during which detailed information about the study and the app was provided. If the adolescent remained interested, the consent form was reviewed, and pre-measurement (T0) appointment was scheduled, allowing a minimum of seven days for consideration. Written informed consent was obtained from the adolescent and, when applicable, from a parent or legal guardian for participants under 16 years of age.

Following informed consent, T0 was conducted, and the M-path app was installed to administer daily questionnaires. Participants received daily reminders to complete these questionnaires throughout the 2-week baseline phase, the 4-week intervention phase, and the 2-week follow-up phase. After completion of the baseline phase, participants commenced use of the Feelee app. Upon completion of the intervention phase, the T1 assessment was conducted, during which the Feelee app was uninstalled. The follow-up phase continued for two weeks with daily questionnaire completion, concluding with the T2 assessment and a semi-structured interview. Additionally, a final interview was conducted with the involved clinician. Three months following study completion, participants were re-contacted and invited to participate in a follow-up assessment (T3), which included questionnaire administration and a semi-structured interview.

Participants received a gift voucher, with the total amount based on the number of completed daily questionnaires, extended assessments (T0, T1, T2, and T3) and emoji entries made within the Feelee app. In line with common practice in EMA studies, a small bonus was provided for weeks in which all daily questionnaires and Feelee entries were completed, to support adherence to the intensive data-collection protocol [71,72]. The incentive was meant to compensate for

the time and effort involved in the study; motivation to use the app's techniques was supported within the treatment itself, where clinicians encouraged adolescents to use their Feelee entries to reflect on daily experiences and treatment goals. If participants met all conditions throughout the study period, the total compensation could amount to a maximum of 38 euros. At T0, the researcher and the participant agreed on the preferred frequency of compensation updates, which in most cases resulted in a weekly update.

## Measures and materials

In this section we first provided a more detailed description the Feelee app, the intervention tool being studied. Subsequently, we outlined the primary outcome, emotion regulation, which was assessed through daily repeated measurements. This is followed by an overview of the secondary outcome measures, which were administered at pre-intervention, post-intervention, and follow-up. Finally, the third objective, exploring perceptions of participants and involved clinicians regarding the effects and usage Feelee. This was addressed through semi-structured interviews, for which methodological details are described in the final part of this section. An overview of all measures and corresponding assessment time points is provided in Table 1.

 *Feelee App.* At the start of the intervention phase, participants installed the Feelee app, (version 1.1.0), on their personal devices (iOS or Android). The Feelee app is owned by Garage2020, an innovation network within the youth care sector, and is free to use. In this study, participants received a pre-generated Feelee account linked to a non-personal email address to facilitate a smooth installation process and safeguard their privacy. During the use of Feelee, users are notified three times a day (by default at 8 a.m., 12 p.m., and 9 p.m., although these times can be adjusted) to select an emoji that reflects their current emotional state. This is followed by three brief reflective questions, 'How come you feel this way?', 'What are you doing?', and 'Who are you with?', which can be answered by selecting pictorial options or providing a written response (Fig 3). Furthermore, Feelee collects two types of passively sensed data: daily step count and hours of sleep. Both the self-reported and passively collected data are displayed in a dashboard to provide a comprehensive view of daily patterns and to support reflection on how sleep and activity relate to emotional states.

**Table 1. Measures, instruments and moment of assessment.**

| Objectives | Instrument | T0 | DQ | Intervention | T1 | T2 | T3 |
|---|---|---|---|---|---|---|---|
| **Primary** | | | | | | | |
| Emotion regulation | Selected items DERS-18 RESS-EMA | x | x | | x | x | x |
| **Secondary** | | | | | | | |
| Emotional differentiation | PANAS | x | | | x | x | |
| Emotional awareness | MAIA (subscale) | x | | | x | x | |
| Self-insight and reflection | SRIS-Y | x | | | x | x | |
| Treatment motivation | ATMQ | x | | | x | x | |
| Treatment alliance | WAV-12 | x | | | x | x | |
| **Other** | | | | | | | |
| Demographic information | Questionnaire | x | | | | | |
| Treatment integrity[1] | Questionnaire | | | x | | | |
| Medical file information | File information | x | | | | | |
| Evaluation Feelee app | Semi-structured interview | | | | | x | x |

*Note:* T0: premeasurement; DQ: daily questionnaires; T1: postmeasurement; T2: follow-up measurement; T3: 3-months follow-up measurement. [1] completed by the clinician.

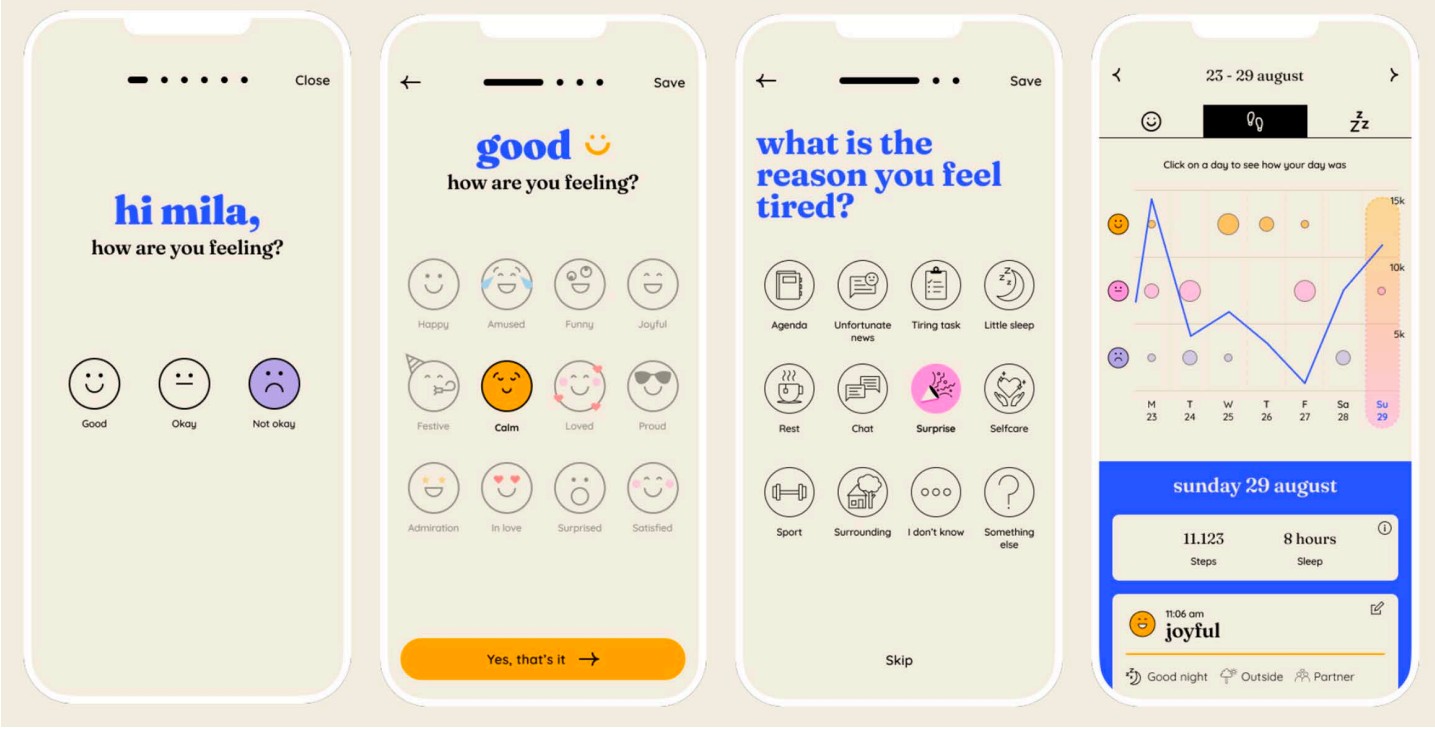

**Fig 3. Screenshots of the Feelee app.** Note: The presented figures serve as illustrative examples and contain no authentic individual data, only pseudonymized information.

During the intervention phase, participants used Feelee independently by entering at least one emoji daily and permitting access to step and sleep data from their phone's health app. In addition, the Feelee data shown in the dashboard were discussed weekly as part of the treatment-as-usual sessions. To enable a proper and consistent discussion of the Feelee data, clinicians received a two-part training prior to study start. In the first session, they were introduced to the app's functionalities and the structure of the data summaries. In the second session, clinicians practiced interpreting and discussing these summaries using their own Feelee entries, allowing them to become familiar with the app's daily routine and the types of patterns adolescents might show. Discussion of the Feelee data were guided by the four-step approach of emotion regulation approached introduced during the training: (1) recognizing emotions via emojis and bodily experience, (2) reflecting on entry moments and context, (3) identifying emotion patterns linked to activities or social factors, and (4) discussing how sleep and activity patterns may affect the emotional state. This approach was also provided in a written handout summarizing the key steps and guidance, which clinicians could consult throughout the study.

***Emotion regulation.*** The primary outcome was assessed through daily questionnaires throughout the ABA design. To facilitate this, the M-path app was used to administer the daily questionnaire. Participants received a notification at a preferred time each day to complete the questionnaire, and if it was not completed, a reminder notification was sent after 120 minutes. To enable daily data measurements, six items were selected to reflect the three phases of emotion regulation targeted by Feelee: recognition, reflection, and management. All items were adapted from the 'Difficulties in Emotional Dysregulation Scale' (DERS-36) [73] or 'Regulation of Emotion Systems Survey for Daily Usage' (RESS-EMA) [74]. To improve suitability for daily assessment, items were shortened and reformulated into single-sentence statements with a 24-hour reference frame. Recognition was assessed with recognition (clarity) and suppression items (e.g., "In the past 24 hours, I had no idea how I was feeling"; "I pretended I wasn't upset"). In this context, suppression was conceptualized as

an automatic, unconscious response occurring early in the emotional process. For reflection, rumination and emotional reappraisal were assessed (e.g., "I thought about the emotional event again and again"; "I looked at the situation from several different angles"). For management, impulse control and distraction were assessed (e.g., "I had difficulty controlling my behaviors"; "I engaged in activities to distract myself"). Responses were provided on a 5-point scale, ranging from 'strongly disagree' to 'strongly agree' (DERS-36) [73] or scale from 0 = not at all to 100 = very much (RESS-EMA) [74]. For recognition (clarity) and manage (impulse) items, scores were recoded to ensure that higher scores consistently reflected higher levels of each construct.

Secondary objectives were assessed at pre-, post-, and follow-up time points to gain a more in-depth understanding of changes in (a) emotional developmental factors and (b) treatment-related factors, both at the individual and group level.

***Emotional differentiation.*** Emotional differentiation was assessed using the Positive and Negative Affect Schedule (PANAS) [75] a 20-item scale that required participants to rate 10 positive and 10 negative emotions on a 5-point Likert scale ranging from 'Not at all' to 'Extremely.'

***Emotional awareness*** was measured using the Multidimensional Assessment of Interoceptive Awareness (MAIA) [76]. In this study, only the 'Emotional Awareness' subscale was administered, which consisted of 32 items across 8 subscales rated on a 6-point scale from 'Never' to 'Always.' Treatment characteristics, including motivation and alliance, were also assessed.

***Self-reflection and insight.*** To assess self-reflection and insight, the Self-Reflection and Insight Scale for Youth (SRIS-Y) [77] was utilized. This 17-item scale was rated on a 6-point Likert scale from 'Strongly disagree' to 'Strongly agree,' and included two subscales: Self-reflection and Insight, each providing separate scores.

***Treatment motivation*** was measured using the Dutch Adolescent Treatment Motivation Questionnaire (ATMQ) [78], a self-report scale with 11 items rated on a 3-point Likert scale: 'Not true,' 'Somewhat true,' and 'True.'

***Therapeutic alliance*** was assessed by the Working Alliance Inventory-12 (WAV-12) [79], a 12-item self-report questionnaire, was used. The WAV-12 evaluated the collaboration between the clinician and the adolescent on a 5-point Likert scale from 'Rarely or never' to 'Always,' with higher scores indicating a stronger therapeutic alliance.

***Treatment integrity*** was assessed using a self-developed questionnaire comprising three multiple-choice questions on whether a treatment session occurred that week, whether Feelee data were available, and whether the data were discussed. One open-ended question allowed for additional comments on the use of Feelee in treatment

The third and final objective aimed to gain an understanding of the use and experiences of adolescents and therapists with the Feelee app, assessed during both follow-up measurements.

***Semi-structured interviews.*** The interviews focused on perceived effects on emotional skills and treatment-related factors, such as motivation and the therapeutic alliance. Participants reflected on observed changes during therapy and the possible role of the app in these developments. Additional topics included the use of other smartphone apps and the integration of data into treatment. The full topic list is included in S1 File.

## Data-analysis

The analysis of the primary objective was divided in two parts. First, we conducted visual inspection analyses on individual level using a *Shiny app* for Single-Case experimental Design [80]. For each participant, a graphical representation was generated to display the scores for each item of the daily emotion regulation questionnaire, connected over time. Participants needed at least five data points per phase to ensure reliable inspection [81]. The visual inspection focused on (a) the variability between data points, (b) changes between phases based on median scores, and (c) trends within and across phases. Group-level changes and trends were summarized using boxplots of item-level score distributions per phase, generated in R Statistical Software (version 4.2.1) [82]. To enhance interpretability, the original 5-point Likert scale responses for the recognition (clarity) and management (impulse control) items were transformed to a 0–10 scale through linear rescaling (i.e., multiplying each value by 2.5). Additionally, randomization tests were conducted in Shiny

apps as well [80]. Cohen's d was calculated for each individual participant and averaged at group level, with effect sizes interpreted as 0.2 (small), 0.5 (medium), and 0.8 (large) [83]. Subsequently, a permutation test was performed [84]. This test included a correction for autocorrelation, ensuring that potential dependencies between daily scores are accounted for. Tests at individual participant level were performed one-sided in the hypothesized direction. Individual $p$-values were obtained by repeatedly randomizing scores between phases and calculating the proportion of differences equal to or exceeding the observed. At group level, the sum of individual $p$-values was compared to the null distribution, using the properties of p-values that they are uniformly distributed between 0 and 1. Last, Tau-U scores were calculated to quantify change magnitude and direction per participant. Unlike the permutation test, Tau-U accounts for undesirable trends within phases, offering a more precise intervention effect estimate [85]. Tau-U values range from −1–1, with effect sizes interpreted as <0.2 (small), 0.2–0.6 (moderate), 0.6–0.8 (large), and >0.8 (very large) [86]. For all statistical tests a type 1 error rate of .05 was used.

Secondary outcome measures were analyzed using IBM SPSS (version 17) and R Statistical Software. For emotion differentiation, sum-scores were calculated separately for both positive affect (PA) and negative affect (NA) scales. Paired-sample t-test assessed changes in standard deviation scores for each subscale. For emotional awareness, self-reflection, insight, motivation, and therapeutic alliance, randomization tests compared mean scores between time points (T0, T1, T2) using $t$-statistics. The observed $t$ was evaluated against a reference distribution from randomized data. The $p$-value reflected the proportion of more extreme randomized $t$-values. Individual changes were also assessed with the Reliable Change Index (RCI), calculated as the difference between time points divided by the standard error [87]. An RCI-score of ±1.96 indicates statistically reliable change, suggesting a meaningful change [88].

The semi-structured interviews were recorded and transcribed. One participant preferred not to be recorded. Therefore, a summary report of the interview was made. The transcripts and this report were independently coded by two researchers using MAXQDA (version 24). A reflexive thematic analysis was applied [89,90]. The interview topic lists offered starting points for the analysis (S1 File). These topics served as sensitizing concepts rather than fixed themes. During the first coding round, additional patterns emerged inductively. Emerging codes and interpretations were discussed within the research team in a reflexive dialogue to explore alternative readings and sharpen interpretations. In cases of uncertainty or disagreement, a third researcher was consulted. In the second coding round, the initial codes were revisited, refined and regrouped. Codes were merged, split or rephrased where needed, and connections between codes were explored to see how more coherent preliminary themes could take shape. This iterative process resulted in three overarching themes: (1) general experiences with Feelee, (2) effects of Feelee on emotion regulation, and (3) working elements of Feelee in treatment. Each overarching theme consisted of several subthemes, which were grounded in clusters of related codes.

### Ethical considerations

This research project was approved in April 2023 by the independent Medical Ethics Committee of the Vrije Universiteit Medical Center (reference number: 2022.0398). As the study involves a clinical application of a smartphone as an integrated part of treatment, the Medical Device Regulation (MDR) applies. Therefore, the study was evaluated not only on ethical aspects but also on technical functionality and safety. Both the research data and the Feelee data were processed and stored in accordance with the General Data Protection Regulation (GDPR). Furthermore, together with the privacy officer of Amsterdam UMC, a Data Protection Impact Assessment (DPIA) and an evaluation of confidentiality, integrity, and availability (CIA) were conducted to identify potential risks and measures for data collection and storage.

### Results

The results are presented at the group level to provide a structured and comprehensive overview of the observed trends and patterns. Rather than presenting individual case outcomes in detail, the findings focus on overarching developments

in the primary and secondary outcome measures. In addition, key themes from the semi-structured interviews are presented, capturing how adolescents and clinicians experienced the use of the app in clinical practice. More detailed results on individual cases can be found in S2 File. Furthermore, to illustrate the heterogeneity of individual trajectories, three different cases are presented in more detail in S3, S4 and S5 Files. Each case represents a markedly different pattern of Feelee usage and change, reflecting high, moderate, and minimal improvement on the emotion regulation items during the intervention phase. Together, these cases illustrate variation in both timing and extent of observed effects, offering a more nuanced perspective on individual app usage and effects in clinical practice. The names that are used in these individual cases are pseudonymized.

## Descriptive characteristics

The majority of 22 participants were male (86%), which is characteristic of the forensic population, where approximately 80% of adolescents in care are male [91]. Notably, all participants (100%) had parents who were either divorced or separated. Most participants (59.5%) resided permanently with one of their parents. Additionally, 68% of participants reported having been convicted by a court of law, while the remaining 32% had not been convicted but were receiving treatment as prevention. The participants were provided with various forms of therapeutic intervention, which are described in more detail in Table 2.

## Primary outcome

First, cases were inspected on the presence of a sufficient number of data points (i.e., more than five) in each study phase. Among the participants who dropped out, one individual who discontinued participation during the follow-up phase had sufficient data points in the preceding phases and was therefore retained in the analyses. One participant who completed all phases was excluded due to insufficient number of data points. Consequently, 14 participants were included in the analyses, with completed data across all phases available for 13 participants. Furthermore, study progress and treatment integrity were evaluated based on clinician reports during the intervention phase. For one participant (7%), the app was discussed weekly. For 5 participants (36%), Feelee data were addressed in 2–3 of the 4 treatment sessions. For 6 participants (43%) the Feelee data was discussed once and for 2 participants not all (14%). A detailed overview of the study process for each participant, including information on the frequency of Feelee data discussions, is provided in S2 File. No serious or adverse events occurred during the study.

To access changes in the emotion regulation across study phases, visual inspections and randomization tests were conducted. Fig 4 presents a boxplots of emotion regulation items for each phase, including both individual and group means scores. Table 3 shows the corresponding effect sizes, p-values, and the median of individual Tau-U scores for each phase comparison. Full results for all individual participants are provided in S1 File.

**Recognition.** For the first step of emotion regulation, recognition (clarity), an increase was expected during intervention and follow-up compared to baseline. Visual inspection of the boxplot (Fig 4) showed an increase between baseline and intervention, followed by a decrease during follow-up. The randomization test, presented in Table 3, revealed a significant increase between baseline and intervention ($p = .002$). Besides the randomization test, Tau-U (Table 3) indicated small non-overlap effects between baseline and intervention for 6 participants, persisting in two during follow-up, suggesting significant change for 6 participants (43%) between baseline and intervention. On the second item, suppression, a decrease in scores was hypnotized and supported by Fig 4. However, randomization test (Table 3) showed no significant effects from baseline to intervention and follow-up. According to Tau-U scores, 4 participants noted a significant small non-overlapping effect between baseline and intervention. Effects between intervention and follow-up remained limited.

**Reflection.** Regarding the reflection items, comprehension (rumination) and comprehension (reappraisal), an increase was expected starting at intervention and continuing during follow-up. Contrary to this, Fig 4 showed decreases during all

**Table 2. Demographic characteristics.**

| Demographic characteristics | N = 22 |
|---|---|
| **Age (mean)** | 17,8 |
| **Gender (%)** | |
| Male | 19 (86%) |
| Female | 3 (14%) |
| **Migration background** | |
| First-generation migration background | 4 (18%) |
| Second generation migration background | 7 (32%) |
| Dutch background | 11 (50%) |
| **Relation status parents** | |
| Separated or divorced | 22 (100%) |
| **Living situation** | |
| Independently | 3 (14%) |
| Alternating between mother and father | 3 (14%) |
| With father or mother | 13 (59,5%) |
| With a relative | 1 (4,5%) |
| Supported housing | 2 (9%) |
| **Daily activities** | |
| Non | 4 (18%) |
| School, specifically | 17 (77%) |
| *• Secondary school* | *4 (18%)* |
| *• Vocational education* | *10 (45%)* |
| *• University of applied sciences* | *1 (4,5%)* |
| *• Other* | *2 (9%)* |
| Employed | 1 (4,5%) |
| **Convicted by a judge** | |
| Yes | 15 (68%) |
| No | 7 (32%) |
| **Treatment approach** | |
| Family therapy combined with individual therapy | 5 (23%) |
| *• Relational Family Therapy (RFT)* | *2 (9%)* |
| *• Systemic Family Therapy* | *3 (14%)* |
| Individual Therapy | 15 (68%) |
| *• Aggression Regulation Therapy* | *2 (9%)* |
| *• Cognitive Behavioral Therapy (CBT) (focus on enhancing emotion regulation skills)* | *5 (23%)* |
| *• Offense analysis* | *3 (14%)* |
| *• Social Support* | *3 (14%)* |
| *• Trauma therapy (TF-CBT)* | *2 (9%)* |
| Unknown | 2 (9%) |

phases. On both items, no significant differences were found between baseline and intervention or follow-up. However, reversed tests indicated significant reductions between baseline and intervention ($p < .001$) and between baseline and follow-up (p = .002), indicating participants reported lower engagement on both reflection items. Tau-U scores showed strong non-overlap effects for 3 participants (21%) on comprehension (rumination) and small effects for comprehension (reappraisal) in 5 participants (36%) between baseline and intervention, with limited follow-up continuation.

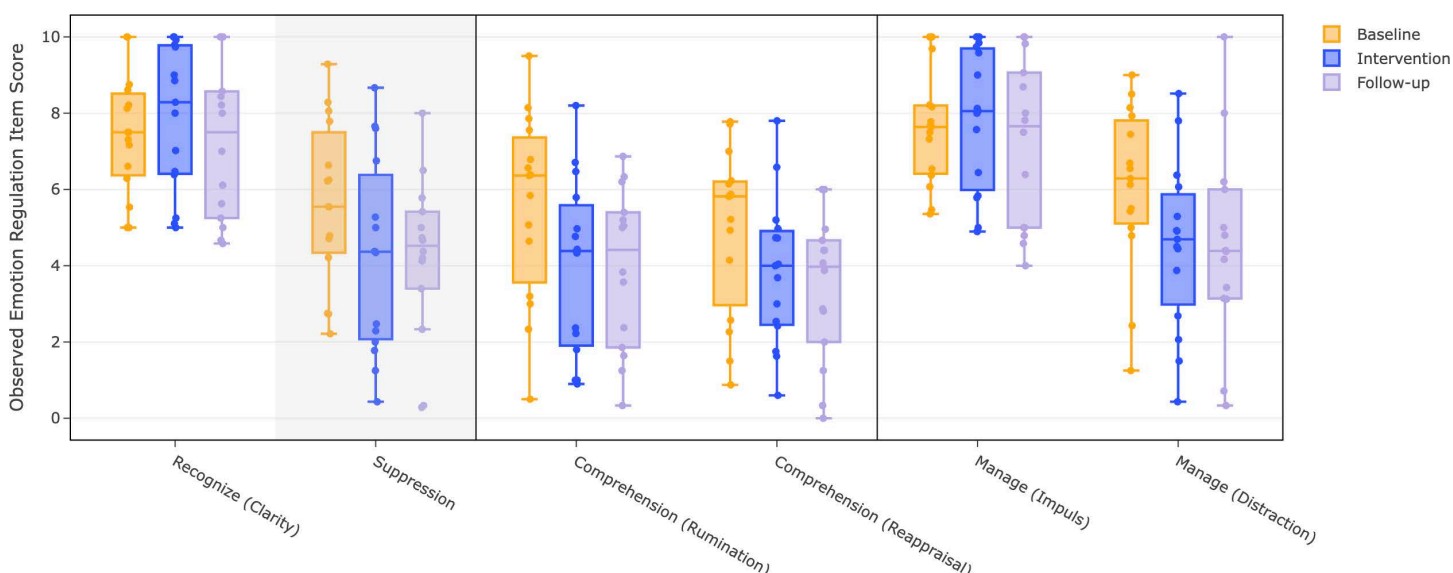

**Fig 4. Boxplot for emotion regulation items for each phase.** *Note.* The soft grey background of 'suppression' indicates an anticipated decrease. For all other items, an increase was expected.

**Managing emotions.** For impulse control, an increase was expected throughout study phases. Both Fig 4 and Table 3 confirmed this pattern. However, randomization test revealed no significant differences between phases. Tau-U scores did reveal a small but significant non-overlap effect between baseline and intervention for 4 participants (29%), persisting for 1 participant (7%) during follow-up. For distraction, an increase was expected starting in the intervention phase, continuing during follow-up. Instead, Fig 4 shows a notable decrease between baseline and intervention, followed by a slight increase at follow-up. Randomization test confirmed no significant effects in the hypothesized direction, but reversed tests showed significant decreases from baseline to intervention ($p < .001$) and follow-up ($p = .001$), indicating that participants noted less distraction throughout study. Tau-U scores showed small non-overlap effect for 6 participants (43%) with no further significant effects during follow-up.

### Secondary outcomes

We evaluated group differences on both (a) emotional development and (b) treatment factors across pre-, post-, and follow-up measurements.

**Emotional developmental factors.** For emotional differentiation, we aimed to investigate whether there was an increase in variability in responses related to positive and negative affect. To assess this, we performed a paired-sample t-test based on the standard deviation (SD) to explore an increase in emotional variability over time. As shown in Table 4, the variability in positive affect did not significantly increase between T0 and T1, but a significant increase was shown between T1 and T2. In contrast, the variability in negative affect remained relatively stable across all three time points, with no significant differences observed between any of the measurement moments.

Next, we performed randomization tests for emotional awareness and both self-reflection insight subscales. As shown in Table 5, no significant changes were found for emotional awareness. RCI-scores showed a reliable increase in 1 participant between T1 and T2, no change between T0 and T2, and a reliable decrease in 1 participant between T0 and T1. For self-reflection, no significant increases were observed between T0 and T1 or T0 and T2, however, a significant increase occurred between T1 and T2. Reliable Change Index (RCI) scores indicated that 8 participants showed a reliable increase between T0

**Table 3. Effect sizes (Cohen's d), p-values and number of significant cases from normal permutation test, and median individual TAU-U scores for each phase comparison.**

| | N | Expected direction | Cohen's *d* | *p*-value (median) | Significant Cases | Tau-U[1] (group) | Significant Tau-U Cases |
|---|---|---|---|---|---|---|---|
| **Recognize (Clarity)** | | | | | | | |
| Baseline – intervention | 14 | (B<I) | −0.28 | 0.02* | 2 | −0.02 | 6 |
| Intervention – follow-up | 13 | (I<F) | 0.252 | 0.33 | 1 | 0.2 | 2 |
| Baseline – follow-up | 13 | (B<I) | −0.098 | 0.33 | 1 | 0.2 | 2 |
| **Suppression** | | | | | | | |
| Baseline – intervention | 14 | (B>I) | 0.584 | 0.99 | 1 | −0.165 | 4 |
| Intervention – follow-up | 13 | (I>F) | 0.075 | 0.99 | 1 | 0.01 | 1 |
| Baseline – follow-up | 13 | (B>I) | 0.535 | 0.99 | 1 | −0.09 | 3 |
| **Comprehension (Rumination)** | | | | | | | |
| Baseline – intervention | 14 | (B<I) | 0.693 | >0.99 | 0 | −0.215 | 6 |
| Intervention – follow-up | 13 | (I<F) | 0.049 | >0.99 | 1 | −0.02 | 2 |
| Baseline – follow-up | 13 | (B<F) | 0.614 | >0.99 | 1 | −0.25 | 5 |
| **Comprehension (Reappraisal)** | | | | | | | |
| Baseline – intervention | 14 | (B<I) | 0.401 | 0.98 | 1 | −0.06 | 5 |
| Intervention – follow-up | 13 | (I<F) | 0.293 | 0.99 | 1 | −0.19 | 2 |
| Baseline – follow-up | 13 | (B<F) | 0.401 | 0.99 | 1 | −0.24 | 6 |
| **Manage (Impulse)** | | | | | | | |
| Baseline – intervention | 14 | (B<I) | −0.107 | 0.21 | 0 | 0.095 | 4 |
| Intervention – follow-up | 13 | (I<F) | 0.361 | 0.18 | 1 | 0.3 | 4 |
| Baseline – follow-up | 13 | (B<I) | 0.057 | 0.18 | 1 | 0.16 | 4 |
| **Manage (Distraction)** | | | | | | | |
| Baseline – intervention | 14 | (B<I) | 0.623 | >0.99 | 0 | −0.24 | 6 |
| Intervention – follow-up | 13 | (I<F) | −0.017 | >0.99 | 0 | 0.1 | 1 |
| Baseline – follow-up | 13 | (B<F) | 0.644 | >0.99 | 0 | −0.09 | 3 |

*Note.* Individual TAU-U scores are presented in S1 File.

*Indicates p<0.05.

[1]TAU-U group-scores are based on median scores from total participant (N).

**Table 4. Changes in emotional differentiation across time points: Within-person SD scores and paired-sample t-test results for positive and negative affect.**

| | Time point | N | Mean SD | SD | *t*-value (*t*) | df | p-value (p) | Cohens'd | Significant RCI + | Significant RCI − |
|---|---|---|---|---|---|---|---|---|---|---|
| **Positive affect** | T0 | 15 | 1.03 | 0.34 | | | | | | |
| | T1 | 14 | 1.12 | 0.30 | −0.845 | 14 | 0.412 | 0.403 | 9 | 1 |
| | T2 | 14 | 0.82 | 0.38 | 3.351 | 13 | 0.005* | 0.323 | 2 | 5 |
| **Negative affect** | T0 | 15 | 0.88 | 0.43 | | | | | | |
| | T1 | 14 | 0.91 | 0.38 | −0.295 | 14 | 0.773 | 0.396 | 3 | 3 |
| | T2 | 14 | 1.15 | 0.40 | −1.788 | 13 | 0.097 | 0.413 | 5 | 1 |

*Note.* T0=pretreatment; T1=posttreatment; T2=follow-up. Subscales based on total scores.

*Indicates p<0.05.

and T1, while 4 showed a reliable decrease; this pattern was similar across T0–T2 and T1–T2. Last, for insight, randomization tests showed no significant changes at any time point. RCI-scores revealed reliable increases for 3 participants between T0 and T1, 4 between T0 and T2, and 5 between T1 and T2, with similar numbers showing reliable decreases.

**Table 5. Results of randomization test and significant Reliable Change Index (RCI) cases for secondary emotional outcomes.**

| | Time point | N | Mean | SD | p-value (p) | Cohens'd | Significant RCI + | Significant RCI - |
|---|---|---|---|---|---|---|---|---|
| **Emotional awareness** | T0-T1 | 15 | 16.32 | 5.31 | 0.282 | 0.23 | 0 | 1 |
| | T1-T2 | 14 | 18.07 | 5.43 | 0.047* | −0.51 | 1 | 0 |
| | T0-T2 | 14 | 14.87 | 7.95 | 0.118 | −0.35 | 0 | 0 |
| **Self-reflection** | T0-T1 | 15 | 41.05 | 6.39 | 0.325 | 0.02 | 8 | 4 |
| | T1-T2 | 14 | 43.57 | 8.01 | 0.044* | −0.51 | 8 | 3 |
| | T0-T2 | 14 | 40.13 | 6.45 | 0.067 | −0.44 | 8 | 2 |
| **Insight** | T0-T1 | 15 | 23.64 | 6.28 | 0.328 | 0.1 | 3 | 3 |
| | T1-T2 | 14 | 22.86 | 3.61 | 0.102 | −0.37 | 5 | 4 |
| | T0-T2 | 14 | 22.13 | 5.54 | 0.218 | −0.22 | 4 | 3 |

*Note.* T0 = pretreatment; T1 = posttreatment; T2 = follow-up. Self-reflection and Insight were measured using sum-scores. Emotional awareness is based on the subscale mean score.

*Indicates $p < 0.05$.

**Treatment factors.** Randomization tests assessed also changes in treatment motivation and treatment alliance across the T0, T1, and T2 time points. As shown in Table 6, no significant changes were found in treatment motivation over time. Additionally, the Reliable Change Index (RCI) analyses did not indicate any reliable improvements. In contrast, treatment alliance showed a significant increase between T0 and T2. RCI-scores revealed a reliable improvement for 3 participants from T0 to T1, 2 participants from T0 to T2, and 3 participants from T1 to T2. Conversely, a slightly higher number of participants showed a reliable decrease: 4 between T0 and T1, 5 between T0 and T2, and 2 between T1 and T2.

## Qualitative results

Qualitatively, the result focus on the perspectives of both adolescents and clinicians regarding their experiences with Feelee, its perceived effects and working elements in treatment. A total of 31 interviews were conducted: n = 14 interviews with adolescents and n = 11 with clinicians at the first follow-up, and an additional n = 6 interviews with adolescents three months later. An overview of the themes and responses from both adolescents and clinicians are presented in Table 7.

**Experiences with Feelee.** First, regarding experiences of using the Feelee app, most adolescents reported being enthusiastic (n = 10). As one adolescent described: "*It was really easy, it actually blended quite well into my routine. (…) How I was feeling, or if something was bothering me, I could easily put it in.*" (A19). Two adolescents were more critical, citing forgetfulness (n = 1) or perceiving the app as childish (n = 1): "*I always felt… I think it's a bit (…) kind of childish or something.*" (A5). Clinicians were also positive. Several found the app easy to use (n = 4) and suitable for integrating into treatment (n = 2), particularly to support emotional reflection: "*I think the Feelee app is a really easy and accessible way*

**Table 6. Results of randomization test and significant Reliable Change Index (RCI) cases for secondary treatment outcomes.**

| | Time point | N | Mean | SD | p-value (p) | Cohens'd | Significant RCI + | Significant RCI - |
|---|---|---|---|---|---|---|---|---|
| **Treatment motivation** | T0-T1 | 15 | 24.95 | 3.15 | 0.112 | −0.42 | 0 | 0 |
| | T1-T2 | 14 | 27.14 | 4.66 | 0.512 | 0 | 0 | 0 |
| | T0-T2 | 14 | 26.80 | 3.49 | 0.17 | −0.26 | 0 | 0 |
| **Treatment alliance** | T0-T1 | 15 | 48.50 | 8.97 | 0.067 | 0.4 | 3 | 4 |
| | T1-T2 | 14 | 47.57 | 10.53 | 0.354 | −0.1 | 3 | 2 |
| | T0-T2 | 14 | 45.80 | 11.87 | 0.034* | 0.55 | 2 | 5 |

*Note:* T0 = pretreatment; T1 = posttreatment; T2 = follow-up. Treatment motivation was calculated using the mean score; treatment alliance is based on the sum-score. * Indicates $p < 0.05$.

**Table 7. Overview of statements from both adolescents and clinicians semi-structured interviews.**

| Theme | Adolescents (*n* = 12) | Clinicians (*n* = 11) |
|---|---|---|
| **General experiences with Feelee** | Positive, helpful in treatment (n = 10) More critical, forget to use it (n = 1), app seen as childish (n = 1) | Positive, easy to use (n = 4) and suitable to integrate in treatment (n = 2) More critical, limited engagement possibilities (n = 2) and passive data issues (n = 5). |
| **Effects of Feelee on emotion regulation** | To think more deeply about emotions (n = 7) | Increased emotional awareness (n = 4) and self-insight (n = 3) |
| | To pause and process emotions (n = 2) | Increased emotional recognition (n = 3) |
| | Emotional release (n = 2) | To think more about positive emotions (n = 2) |
| | No effects experienced (n = 2) | No effects for emotional differentiation (n = 2) |
| **Working elements of Feelee in treatment** | Discussed the data (n = 11), did not discuss it (n = 1) | Discussed according to training guidance (n = 1), in their own way (n = 2), limited discussion (n = 1) |
| | Recall emotions and situations (n = 10) | Insight in daily emotions and activities outside treatment sessions (n = 7) |
| | Expressing emotions by emoji's verbally (n = 3) | Expressing emotions by emoji's verbally (n = 5) |
| | | Supportive to conversation in treatment (n = 9) |

to talk about things, without it always feeling like an extra topic that had to be brought up during the individual sessions." (C7). Criticism focused mainly on limited engagement possibilities (n = 2) and issues with passive data (n = 5). Especially the 3-month follow-up interviews showed a clear decline in usage. In all available interviews, adolescents (n = 6) reported that they no longer used the Feelee app after the study period. One clinician explained the quick decline during the intervention to a lack of interaction: "*The core of the Feelee app is solid (…), but a next step would be being able to view more personalized input over time*." (C2). Regarding the issues with passive data, adolescents reported several problems. Three adolescents indicated that the sleep data did not work properly (n = 3), and for two adolescents it did not register at all (n = 2). Clinicians confirmed this pattern and added that, for two adolescents, the sleep data was not accurate (n = 2). As one clinician explained: "*I heard from adolescents that the sleep data is not yet really accurate.*"(C7), referring to the sleep data not reflecting the adolescents' actual sleep hours.

**Perceived effects of Feelee.** The second main theme involved experiences regarding the effects of Feelee, did a small majority of the adolescents (n = 7) indicated that Feelee helped them to reflect more deeply on their emotions. As one participant described: *"I just started thinking about it more, like: 'Oh, yesterday I felt that way, you know?' I was kind of, uh, low on energy, for example or I was angry or something."* (A9). Clinicians emphasized that this reflection was more about increased 'awareness' (n = 4) and self-insight (n = 3). One clinician noted: *"I actually noticed that when she started using the app, she became much more aware of what was happening within herself on a daily basis. She also began to think, 'Hey, where is this coming from?'"* (C1). Some adolescents (n = 2) mentioned that Feelee helped them pause and process emotions, while others (n = 2) described it as a tool for emotional release. As one participant explained: *"It's easy for me to put my emotions into an app like that, to remember how I felt on a certain day (…). And if, for example, I can't talk to anyone about it, at least I still have the app."* (A19). While adolescents noted these benefits, clinicians observed an increase in emotion recognition (n = 3): *"It may sound simple, but with him, a lot of things just pass by. (…) It starts with recognizing and acknowledging emotions. I think for this adolescent, just being aware of that is already a big step forward."* (C8). Clinicians also mentioned that Feelee sometimes supported reflection on positive emotions. However, a

few adolescents (n = 2) reported no noticeable emotional effects from using the app. According to clinicians, emotional differentiation remained limited. As one explains: "*It's noticeable that this group is actually quite undifferentiated in their emotions, and that it was still difficult for them to do.*" (C7).

**Working elements in treatment.** Concerning the working elements of Feelee in treatment, the qualitative results first provided a more in-depth understanding of how the Feelee data were discussed during sessions. As shown in the quantitative data, weekly discussion of the app was limited. Although most adolescents stated that they had discussed the Feelee data with their clinician (n = 12), one adolescent who had used the app less frequently explained: "*The Feelee app, no not really. But that was mainly because I didn't use it very often*" (A16). Clinicians also offered further explanation about how the discussion took place. Three clinicians described their approach, for example using the guidance from the training (n = 1) or integrating the app in their own way (n = 2). One clinician reflected that the discussion decreased because other priorities in treatment became more urgent: "*And then it faded into the background, because we were focusing on other things that became more important*" (C2).

When the Feelee data was discussed, both adolescents and clinicians described several working elements that contributed to treatment. Adolescents (n = 10) stated that the app's weekly overview helped them recall emotions and past situations. Clinicians found this valuable for gaining insight into daily emotions and activities outside of treatment sessions. One clinician explained: "*I'm there, yeah, maybe one hour a week or sometimes two. But what someone is doing outside of that, well, you start to get more and more insight into that through an app like Feelee.*" (C6). Both adolescents (n = 3) and clinicians (n = 5) highlighted the usefulness of emojis for expressing emotions verbally during treatment. As one adolescent shared: "*(...) Sometimes I find it hard to explain things. So being able to show it like that in the figure and then reflect on it. That was really helpful.*" (A8). Similarly, a clinician noted: (…) "*When I asked questions like 'Did you see your girlfriend?', he couldn't really express that. But with the Feelee app, (…) he could link it to an emoticon and tell a whole story from beginning to end.*" (C9). Additionally, clinicians found Feelee supportive for monitoring and discussing emotions and related behaviors in treatment. As one clinician said: "It did work in the sense of: "*Oh right, I need to keep an eye on that.' So, I did ask about it. And that was really thanks to the Feelee app that made more alert to it.*" (C6).

## Discussion

The present study aimed to explore the initial effects of Feelee in enhancing emotion regulation skills among adolescents in forensic outpatient care. Feelee was incorporated during a 4-week intervention phase, in addition to treatment as usual. The primary objective focused on improvements in three core components of emotion regulation: recognition, reflection, and management. Secondary objectives included a more in-depth examination of changes in (a) emotional developmental factors and (b) treatment-related factors, assessed at pre-, post-, and follow-up. And the third and final objective involved a qualitative exploration of both adolescents' and clinicians' perspectives on the effects and use of the Feelee app.

Findings on the primary outcome showed a significant increase in emotional recognition. However, contrary to expectations, we observed decreases in the items related to reflection (rumination and reappraisal) and management (distraction). For the secondary outcomes, indicators of improvements emerged during follow-up in positive emotion differentiation, emotional awareness and self-reflection. No clear change was observed in treatment motivation, while therapeutic alliance showed a significant increase over time. Qualitative data confirmed that both participants and clinicians were generally enthusiastic about the use of Feelee in treatment, although they also identified areas for improvement, such as technical challenges and limited integration of app data into therapy sessions. While results are presented at the group level, three illustrative case studies, included in the supplementary materials, offer a more detailed understanding of the heterogeneity in how Feelee was used and experienced across individual participants.

The results on the primarily outcome, emotion regulation, revealed interesting patterns for further discussion. Before discussing these patterns in detail, it is first important to note that the results on the primary outcome, emotion regulation, should be interpreted in light of the exploratory nature of the analyses and the multiplicity of statistical comparisons

inherent to multi-outcome SCED research. Although this increases the nominal risk of Type I error, the study prioritized visual inspection and TAU-based effect size patterns over isolated significance testing, consistent with current SCED methodological recommendations [67–69].

Regarding emotional recognition, we found a significant increase in emotional recognition on group-level after participants were introduced to the Feelee app. This within-person change between baseline and intervention suggests that Feelee supported these adolescents in recognizing their emotions more accurately. Clinicians reported similar impressions, noting that adolescents more often identified and expressed their emotions during sessions. Although these results are exploratory, they offer an initial indication that Feelee was supportive in practicing to recognize their emotions more accurately. Furthermore, the results suggest that even a short-term, four-week period of structured and focused emotional training through a low-threshold tool such as Feelee may help adolescents strengthen emotional recognition as the first crucial step in emotion regulation. These initial findings may first support the idea from mHealth research that daily smartphone check-ins offer adolescents opportunities to practice emotional skills in their own everyday environment and may therefore complement traditional care in a meaningful way [49,92,93]. This interpretation also aligns with earlier work in young offenders, where a brief structured training led to measurable improvements in recognizing negative emotions [94]. Strengthening recognition is considered important because greater clarity of emotional states can support later regulatory processes, particularly in adolescents who experience difficulties with emotional insight [94]. However, follow-up measurements showed that this improvement did not continue after the intervention ended, which indicates that the effect was limited to the active use of the app and suggests that more sustained practice may be necessary.

Further noteworthy pattern in the primary outcome emerged for both reflection items: rumination and reappraisal. While we initially expected that Feelee would enhance participants' reflection, the reversed analysis showed a significant decrease in self-reported rumination and reappraisal after the start of the intervention. This finding raises several considerations. It may be linked to the emotional capacities of adolescents in forensic care [21,23]. Participants primarily showed improvements in emotion recognition, an initial stage of emotion regulation. However, reflection is a higher-order cognitive skill that likely requires more practice than the intended four-week intervention period allowed for. In addition, despite pre-testing the daily questionnaire, it remains possible that both reflection items were not measured accurately, potentially introducing biases or leading to misinterpretations. For instance, the item 'rumination' can be interpreted as a reflection on a negative experience instead of a reflection of a potential positive experience [95]. Therefore, a decrease could also indicate less reflection on negative experiences but potentially increase of positive experiences. Similarly, for reappraisal, participants may have interpreted 'the situation' in varying ways [96,97]. These differences may have influenced which types of situations participants reflected on, thereby affecting the consistency and interpretability of their responses. It is plausible that these varying interpretations may have influenced the outcomes.

Beyond the primary outcomes, the secondary measures revealed several relevant outcomes, particularly at follow-up measurements. Adolescents showed significant improvements in positive emotional differentiation, self-reflection, and emotional awareness between post- and follow-measurements. This may indicate that there might be a delayed development of emotional competencies among adolescents, although this interpretation should be treated with caution. Rather than showing immediate effects, the results suggest that Feelee may have initiated changes in these domains during the intervention phase, while actual improvements apparent at a later stage. An alternative explanation is that these constructs were assessed with different questionnaires capturing broader or more gradually developing aspects of emotional functioning, which may naturally show change over longer timescales. These secondary findings should therefore be interpreted as exploratory signals rather than confirm evidence of a possible intervention effect.

Regarding the secondary treatment-related factors, did we not find further improvement in participants' motivation throughout the study. Notably, the high motivation scores at baseline indicate that the adolescents who participated were already motivated at the start of the study. However, this also highlights the complexity of reaching adolescents who are not motivated for treatment, as this group was not targeted in this study [98,99]. For treatment alliance, we

first observed a slight decrease, followed by a significant improvement in participants' reported alliance throughout study. However, this pattern does not automatically imply that the improvement was caused by the use of Feelee. Since adolescents continued their usual treatment during the study, it is well possible that they naturally developed a stronger relationship with their clinician as treatment progressed. Nevertheless, interview data indicated that Feelee supported the therapeutic dialogue by making emotional experiences more concrete and easier to discuss. In this way, these findings align with earlier studies showing that the integration of digital tools in treatment, can facilitate shared goal setting and foster a more shared language between adolescents and clinicians [100]. For adolescents, this enhances feelings of being understood and, conversely, helps clinicians better understand their patients and provide more tailored care [101]. Feelee may therefore be a promising tool to support the therapeutic alliance and help strengthen the engagement in treatment.

A last and important consideration that may have influenced the study findings concerns the treatment integrity observed during the intervention. The treatment log revealed that for the majority of participants, Feelee data was discussed only once or not at all during the 4-week period. The qualitative findings provide additional insight, showing that clinicians often deprioritized discussing the app because other issues in treatment demanded more urgent attention. This pattern is common in forensic outpatient care, where the complexity and multiplicity of adolescents' needs frequently require clinicians to shift focus to acute behavioral, familial or safety-related concerns [63,102]. As a result, integrating new tools such as Feelee into sessions can be challenging, not because of a lack of willingness, but because of these structural and contextual barriers. Strengthening treatment integrity in future studies will likely require clearer implementation guidance, structural support for clinicians and dedicated time within sessions to review app data in a consistent way [103].

## Strengths and limitations

This study has various strengths and limitation. A major strength is that this study belongs to a limited body of research investigating the initial effects of a data-driven smartphone app aimed at enhancing emotional regulation skills among adolescents in forensic outpatient settings. By conducting this study, we contribute to addressing the lack of research on new technologies for adolescents in forensic outpatient settings. Second, although the forensic population is considered complex for research due to typically low engagement and high dropout rates, we were able to recruit and include a sufficient number of participants for the data analyses. Another strength is the use of a mixed-methods design, which enhanced the understanding of study outcomes and their implications for clinical practice. Finally, in addition to the study design, the research was conducted under real-world, uncontrolled circumstances, making the outcomes directly valuable for clinical implementation.

This study has also several limitations. First, although the study design (SCED) incorporated structured baseline, intervention, and follow-up phases, the small overall sample size restricts the generalizability of the findings and calls for caution when drawing conclusions about the app's effects for a broader population. Nevertheless, SCEDs offer particular methodological advantages over traditional randomized controlled trials (RCTs), especially in exploratory research. They allow for detailed insights into individual change processes and contextual factors that often remain unexamined in standard group designs. Second, although recruitment efforts were as broad as possible, it was predominantly highly motivated adolescents who agreed to participate, or who were selected for participation by their caregivers. This self-selection may have introduced bias and could result in an overestimation of the intervention's feasibility or perceived usefulness in the broader forensic population. Third, while Feelee collects various types of data, its integration into treatment remained limited. In practice, only the self-reported emoji-based mood data were actively discussed during therapy sessions. Additionally, technical issues hindered the passive data collection (e.g., activity or sleep metrics) in several cases. For these participants, the intervention was limited to its active components, reducing the scope of its originally intended functionality. Consequently, the digital phenotyping component for these individual cases was based solely on self-reported data.

Finally, the daily repetition of the daily questionnaire may have affected the validity of responses. The high frequency and repetitive nature of the items may have led some participants to respond in a socially desirable manner or to answer quickly without fully considering the content of each question. Such tendencies could reduce the accuracy of the data and reflect decreased engagement over time, especially in a population that may have limited tolerance for structured self-monitoring procedures [104].

### Implications for research and clinical practice

The findings also serve several implications for research and clinical practice. First, more larger-scale studies are needed to build on these initial findings, with particular focus on whether more advanced emotion regulation skills may develop after prolonged use (> 4 weeks). It is also important to consider how research designs can be made more inclusive, particularly in forensic populations, where adolescents are often less willing to participate in research. Although low motivation should not be seen as a target in itself, this subgroup is frequently underrepresented in studies, while they may benefit substantially from low-threshold, digitally supported interventions. Future efforts should therefore explore how research procedures and tools can be adapted to better engage these adolescents, for instance by simplifying participation and increasing the accessibility and relevance of research activities. Future efforts should therefore explore how research procedures and tools can be adapted to better engage these adolescents, for example by simplifying participation and increasing the accessibility and relevance of research activities. Close collaboration with adolescents and clinicians throughout each step of the research process is essential, as this not only strengthens engagement but also fosters a shared language and understanding that more closely resonates with adolescents' everyday experiences.

For clinical practice, Feelee may offer added value within treatment and may also hold potential as an initial screening tool for emotional functioning. In this study, the Feelee data provided clinicians with valuable insights into the emotional functioning of adolescents, also outside treatment sessions, helping them guide decisions regarding the focus and intensity of treatment. Furthermore, our findings emphasize the importance of integrating app data into the therapeutic dialogue. Adolescents reported that reviewing their Feelee data during therapy helped them better understand their emotional patterns. However, treatment integrity showed limited implementation of discussing Feelee data within treatment. We therefore advise to actively incorporate these data into sessions, rather than treating the app as a stand-alone tool. This also highlights the need for further clinical guidance on the integration of a data-driven smartphone app, including how to effectively discuss and apply the Feelee data in treatment. At the same time, this should always be done with sufficient flexibility to respond to the complexity of treatment within this population.

### Conclusion

This study conducted a first comprehensive evaluation of smartphone-data driven mHealth app (Feelee) to enhance emotion regulation skills in forensic outpatient treatment. The findings suggest that a 4-week period of using Feelee may be valuable in strengthening emotional recognition. For other steps of emotion regulation, such as reflection and managing emotions, longer periods of use and additional therapeutic support may be required. Regarding treatment outcomes, participant motivation remained stable, whereas the therapeutic alliance significantly improved. Qualitative findings supported these quantitative results, while also indicating that technical issues and limited integration of Feelee data into treatment discussions may have constrained the app's full potential. The heterogeneity in outcomes across the three cases further underscores the effects and functioning of Feelee may differ depending on individual needs and context. Future research should examine whether prolonged use of Feelee can foster the development of potential higher-order emotion regulation skills, such as reflection and more advanced emotional management. To support this, it is essential to involve both adolescents and clinicians in the design and evaluation process, in order to reach a broader population and to develop clear guidance on how to meaningfully integrate and discuss Feelee data within treatment.

## Supporting information

**S1 File. Topic lists semi-structured interviews.**
(DOCX)

**S2 File. Detailed results outcomes all participants.**
(DOCX)

**S3 File. Detailed result outcomes of case 1.**
(DOCX)

**S4 File. Detailed result outcomes of case 2.**
(DOCX)

**S5 File. Detailed result outcomes of case 3.**
(DOCX)

## Acknowledgments

The authors would like to thank all adolescents and clinicians who participated in this study. We are also grateful to the clinicians who were not directly involved in the study but contributed to the recruitment process. Furthermore, we thank Mignonne Curiel, Aaliyah Hansen, Anne Kok, and Imara Semeijn for their valuable support during the preparation, recruitment, data collection, and data analysis phases of the study.

## Author contributions

**Conceptualization:** Merel M.L. Leijse, Levi van Dam, Samantha Bouwmeester, Thimo M. van der Pol, René Breuk, Arne Popma.

**Data curation:** Merel M.L. Leijse.

**Formal analysis:** Merel M.L. Leijse, Samantha Bouwmeester.

**Investigation:** Merel M.L. Leijse.

**Methodology:** Merel M.L. Leijse, Levi van Dam, Samantha Bouwmeester.

**Project administration:** Merel M.L. Leijse.

**Supervision:** Levi van Dam, Samantha Bouwmeester, Thimo M. van der Pol, René Breuk, Arne Popma.

**Validation:** Samantha Bouwmeester.

**Visualization:** Samantha Bouwmeester.

**Writing – original draft:** Merel M.L. Leijse.

**Writing – review & editing:** Levi van Dam, Samantha Bouwmeester, Thimo M. van der Pol, René Breuk, Arne Popma.

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
