## [Decision Letter · Decision Letter 0]

30 Oct 2025

PONE-D-25-44103‘I Gotta Feelee-ing’: Exploring the Effects of a Smartphone app (Feelee) to Enhance Adolescents’ Emotion Regulation in Forensic Outpatient Settings. A Multiple Single-Case Experimental Design.PLOS ONE

Dear Dr. Leijse,

Thank you for submitting your manuscript to PLOS ONE. After careful consideration, we feel that it has merit but does not fully meet PLOS ONE’s publication criteria as it currently stands. Therefore, we invite you to submit a revised version of the manuscript that addresses the points raised during the review process.

Please carefully revise your paper accordingly because one of the reviewers did not recommend your paper but I still think your paper is inspiring. Thanks.  Please submit your revised manuscript by Dec 14 2025 11:59PM. If you will need more time than this to complete your revisions, please reply to this message or contact the journal office at plosone@plos.org . Please include the following items when submitting your revised manuscript:

We look forward to receiving your revised manuscript.

Kind regards,

Mu-Hong Chen, M.D., Ph.D.

Academic Editor

PLOS ONE

Journal Requirements:

For the design and conduct of the study described in this paper, the first author received funding from Quality Forensic Care Youth (‘Kwaliteit Forensische Zorg Jeugd’ in Dutch). Website: https://kfzjeugd.nl/

I have read the journal's policy and the authors of this manuscript have the following competing interests: LvD is an employee of Garage2020, the foundation that provides the Feelee app.

4. We noted in your submission details that a portion of your manuscript may have been presented or published elsewhere. Please clarify whether this publication was peer-reviewed and formally published. If this work was previously peer-reviewed and published, in the cover letter please provide the reason that this work does not constitute dual publication and should be included in the current manuscript.

5. We note that you have indicated that there are restrictions to data sharing for this study. For studies involving human research participant data or other sensitive data, we encourage authors to share de-identified or anonymized data. However, when data cannot be publicly shared for ethical reasons, we allow authors to make their data sets available upon request. For information on unacceptable data access restrictions, please see http://journals.plos.org/plosone/s/data-availability#loc-unacceptable-data-access-restrictions.

Additional Editor Comments:

Please carefully revise your paper accordingly because one of the reviewers did not recommend your paper but I still think your paper is inspiring. Thanks.

Reviewers' comments:

Reviewer's Responses to Questions

**Comments to the Author**

1. Is the manuscript technically sound, and do the data support the conclusions?

Reviewer #1: Yes

Reviewer #2: Partly

Reviewer #3: Partly

Reviewer #4: No

2. Has the statistical analysis been performed appropriately and rigorously? 

Reviewer #1: Yes

Reviewer #2: I Don't Know

Reviewer #3: Yes

Reviewer #4: No

3. Have the authors made all data underlying the findings in their manuscript fully available?

Reviewer #1: Yes

Reviewer #2: No

Reviewer #3: No

Reviewer #4: Yes

4. Is the manuscript presented in an intelligible fashion and written in standard English?

Reviewer #1: Yes

Reviewer #2: Yes

Reviewer #3: Yes

Reviewer #4: Yes

5. Review Comments to the Author

Reviewer #1: Dear Editor,

I write to submit my review comments on the manuscript titled “‘I Gotta Feelee-ing’: Exploring the Effects of a Smartphone app (Feelee) to Enhance Adolescents’ Emotion Regulation in Forensic Outpatient Settings. A Multiple Single-Case Experimental Design”.

The study utilized a mixed-method approach to assess the effect of adding the Feelee app to the usual treatment, aiming to enhance emotion regulation skills among forensic outpatients. It evaluated the effects of Feelee on three core components of emotion regulation targeted by the app: emotional recognition, reflection, and management, as measured daily. The study also examined changes in (a) emotional developmental factors, including emotional differentiation, awareness, self-reflection and self-insight, and (b) treatment-related factors, such as treatment motivation and therapeutic alliance. The study finally used qualitative methods to explore both adolescents’ and clinicians’ experiences with the use and integration of Feelee in therapeutic practice

Here are my comments

General comment: The study diligently followed the Single-Case Reporting Guideline in Behavioural Interventions (SCRIBE) and provided justification for using a non-randomized multiple Single Case Experimental Design (SCED)

1. Authors must provide justification or rationale for choosing a 2-week baseline (A1), a 4-week intervention (B), and a 2-week follow-up phase (A2). What informed the duration of the intervention period, and what would have been the implications if the intervention period had been extended or reduced? Kindly provide justification, which may include references to similar studies, etc

2. The authors stated that three months after the initial follow-up measurement, a long-term follow-up assessment (T3) was conducted to evaluate the potential sustained effects of Feelee. What do you define as long-term? 6 months, 1 year, etc. Kindly clarify in the main document

3. What informed the selection of participants from outpatient forensic teams at two mental healthcare institutions in the Netherlands? Are they the only mental health institutions in the Netherlands? What informed the author's choice of selecting Levvel and Inforsa and not any other mental health facilities? How did you arrive at only two facilities and not any other number? No power analysis? Clarity must be provided in the manuscript

4. Kindly explain what you mean by “0–10” scale through linear rescaling” in the manuscript

5. Yes, the permutation test does not correct for autocorrelation, but how did you assume that the scores on the six daily items do not exhibit substantial autoregression without any rigorous statistical test? Additionally, repeated responses in this case scenario are naturally autocorrelated and must be accounted for in the statistical analyses.

6. I am wondering how applying a correction for autocorrelation could therefore result in spurious effects. The autocorrelation problem affects standard error estimation and the corresponding p-value estimates. Can you show the results, both autocorrelation corrected and uncorrected, as part of the sensitivity analysis?

7. Kindly label the y-axis of the box-whisker plot

8. The primary and secondary outcome measures have been clearly defined, and the authors included a measurement scale for each outcome measure. The intervention has been well described in the manuscript. The results are valid as they followed a standard statistical analytic approach

Reviewer #2: This manuscript presents a mixed-methods evaluation of Feelee, a smartphone application designed to support emotional regulation among adolescents in a forensic outpatient setting. The app collects both self-reported mood data and passive sensor data. However, several revisions are needed to strengthen the manuscript before it can be considered for publication.

Major Comments:

1. The introduction would benefit from a more comprehensive review of the current state of the art. The statement that “current interventions in forensic outpatient settings primarily aim to address the multiproblem profile by focusing on reducing risk factors and increasing protective factors” lacks sufficient context. It would strengthen the introduction to elaborate on what these current interventions typically entail, including examples of evidence-based or widely adopted practices.

2. To strengthen the rationale for the Feelee app, a more detailed review of existing digital mental health tools for adolescents is needed, particularly those incorporating ecological momentary assessment (EMA) or sensor-based monitoring. This would better situate the current study within the broader landscape of digital health innovations and help justify the unique contribution of Feelee in this setting.

3. The study includes participants aged 12 to 23. The manuscript should provide a rationale for this broad age range and clarify how developmental differences were accounted for in the analysis. Were any age-related differences observed in engagement or outcomes? If so, these should be explicitly reported.

4. Some context regarding smartphone ownership, access, and usage habits among this population would help support the feasibility and relevance of the intervention. Is there existing data on smartphone use within forensic adolescent population and were there any barriers to technology use observed during the study?

5. The mention of the Feelee app in the introduction appears premature and somewhat disconnected from the rest of the introduction. Its inclusion would be more appropriate in the discussion section, where its relevance can be evaluated in light of the findings.

6. While it is noted that passive sensor data (e.g., sleep and number of steps) are collected via the Feelee app, a comprehensive list of all data types gathered passively should be provided and the advantages of collecting such data

7. the term "active data" may be confusing in the context of smartphone and sensor-based research. It may be clearer and more accurate to refer to this as "self-report data" to distinguish it from passively collected information.

8. The inclusion of "basic proficiency in smartphone use" as an eligibility criterion warrants clarification. Please explain how this proficiency was assessed, by whom, and whether any standardised or informal measures were used.

9. More detailed information is needed regarding participant compensation. How much were participants compensated? The phrase “agreed frequency of updates” is unclear. Please clarify how and when participants were compensated and whether reminders or incentives were provided.

10. Further information is needed on how data summaries from the Feelee app were generated and shared with participants’ and clinicians. It would be important to outline what guidance or training, if any, was provided to clinicians to support the integration of Feelee app data into their sessions with participants to provide context to its adoption and discussion of data with adolescents.

11. More information is needed about the qualitative component of the study. Specifically, an overview of the interview topic guide would be valuable. This should include the main domains or questions covered in the interviews and the analysis approach, to allow readers to assess the breadth and focus of the qualitative component of the study.

12. The themes presented are quite limited and are very categorical. The authors should distinguish between themes that were pre-defined and those that emerged through inductive thematic analysis. Clarify the approach to coding and provide illustrative quotes for each major theme.

13. How many qualitative interviews were conducted?

14. In the qualitative results section, the term ‘participants’ is used. As there are two participant groups (adolescents and clinicians) it would be advantageous to be explicit as to which group are being referred to.

15. In the results it would be beneficial to separate out how many had their data discussed once and not at all rather than grouping these.

16. Limited discussion of the data is reported. Did participants report any reasons for why it was not discussed?

17. The presentation of the qualitative findings is also very limited and lacks details. The qualitative findings reference issues with passive data collection, but lack specificity. What types of issues were encountered? Providing adolescent or clinician quotes would help illustrate these challenges.

Minor comments

18. The paper could do we further proofreading as grammatical and spelling errors were identified (e.g. missing full stops, hypnotised should be hypothesised)

19. Formatting should be reviewed, particularly the small, disjointed line beneath Table 1.

20. Consider rephrasing “these multi problems” for clarity

21. Some more sub-headings and signposting may improve the readability of the results.

Reviewer #3: The authors investigate the efficacy of the use of a Smartphone App to train emotion regulation in youth in a forensic outpatient setting in a small pilot study. The research question is topical and important, but several comments should be considered.

- Efficacy or effectiveness: while it is true that the authors carry out their pilot study in an outpatient setting involving two clinics, the use of the concept of effectiveness in this preliminary and research setting seems not appropriate.

- Explain the rationale of the timeline of the SCED timeline, why 2-weeks baseline, then 4 weeks intervention and then 2-weeks follow up phase? This seems to be very little time regarding the attempt to influence basic mechanisms such as emotion regulation training

- Exchange the term “psychiatric” conditions with mental disorders, as “psychiatric” describes a profession and mental disorders describe the content of interest

- Were comorbid mental disorders assessed? Please report them

- Participants received monetary incentives: on what grounds did you choose this approach and how do you foster intrinsic motivation to continue using the techniques of the App?

- Steps and “sleep” were assessed. Please indicate more clearly what the goal of this assessment was. Did you really expect that the short-term emotion regulation training would improve sleep and increase physical activity?

- Add examples of the items of DERS and RESS-EMA, and indicate ICCs

- On p. 15, you write that the permutation test, which does not correct for autocorrelation as it is assumed that the six daily items do not exhibit autoregression. This does not seem to be plausible. Please explain

- Please explain why the three cases were chosen to be presented

- Conclusions on the efficacy of the App are overstated. The findings may indicate the way for future research, including larger sample sizes and controlled designs, but interpretation of e.g. distraction values seems a bit too far-reaching. The same holds regarding the interpretation of the secondary effects and the delayed improvement of emotional differentiation and competencies (p.31).

On the other hand, treatment integrity was very low, and this is a major issue that should be discussed in more detail. Why did therapists not engage? Is this a common problem in this patient-therapist group?

Reviewer #4: This manuscript examines the preliminary effects of the Feelee smartphone app as an adjunct to forensic outpatient care for adolescents, using a multiple single-case experimental design combining quantitative and qualitative data. However, several methodological issues substantially limit the interpretability of the findings.

Firstly, the study adopts a multiple single-case experimental ABA design but lacks a well-defined control group, which limits the interpretability of the reported reduction in emotional suppression. It remains unclear whether the reduction in emotional suppression resulted from the Feelee app itself or from concurrent forensic outpatient treatment. Forensic outpatient care inherently includes therapeutic elements that may independently influence emotion regulation. Moreover, adolescents under judicial supervision may show behavioral improvement due to legal or compliance-related factors. As currently designed, the study cannot disentangle the effects of the app from those of the ongoing forensic outpatient program.

Secondly, the manuscript does not provide a rationale or statistical justification for the chosen sample size. No reference to prior effect sizes from studies on emotion suppression or app-based emotion regulation was presented. Although 27 adolescents were recruited, only 14 were included in the main analysis and 13 in the final dataset. The authors should clarify whether this number affords sufficient statistical power to detect the intended effects. Without such justification, the robustness of the reported findings remains uncertain.

Furthermore, the statistical analyses involve numerous comparisons across phases and variables, yet there is no indication that corrections for multiple testing (e.g., Bonferroni) were applied.

In addition to these major concerns, the manuscript provides only minimal information about the nature of the forensic outpatient care provided to participants. It is unclear what kinds of offenses or behavioral problems characterized these adolescents, what psychiatric or psychological conditions they presented with, and what specific treatments (“treatment as usual”) they received. Details such as session content, therapeutic approaches, and treatment duration are essential for evaluating the study context and the plausibility of treatment effects.

Lastly, the qualitative findings mainly describe user experiences with the app, rather than offering explanatory insight into how emotional suppression was reduced. The current qualitative results could be condensed, and integration of qualitative and quantitative findings—either within the discussion or in supplementary materials—would provide a more coherent interpretation of the app’s effects.

In sum, the current design and analytical limitations substantially weaken the evidential value of the findings. Addressing these issues—particularly by adding a proper control condition, justifying sample size, and applying statistical corrections—would be necessary before the results could be considered reliable.

6. PLOS authors have the option to publish the peer review history of their article (what does this mean? ). If published, this will include your full peer review and any attached files.

**Do you want your identity to be public for this peer review?** For information about this choice, including consent withdrawal, please see our Privacy Policy .

Reviewer #1: No

Reviewer #2: No

Reviewer #3: **Yes:** Simone Munsch

Reviewer #4: No

---

## [Author Response · Author response to Decision Letter 1]

16 Dec 2025

We thank the editor and the reviewers for their constructive comments. We have carefully addressed all reviewer and editorial comments in a separate Response to Reviewers document and revised the manuscript accordingly. All changes are indicated in the revised manuscript using track changes.

Thank you for considering our revised submission.

Kind regards,

Merel Leijse

(Corresponding author)

---

## [Decision Letter · Decision Letter 1]

23 Dec 2025

PONE-D-25-44103R1‘I Gotta Feelee-ing’: Exploring the Effects of a Smartphone app (Feelee) to Enhance Adolescents’ Emotion Regulation in Forensic Outpatient Settings. A Multiple Single-Case Experimental Design.PLOS One

Dear Dr. Leijse,

Thank you for submitting your manuscript to PLOS ONE. After careful consideration, we feel that it has merit but does not fully meet PLOS ONE’s publication criteria as it currently stands. Therefore, we invite you to submit a revised version of the manuscript that addresses the points raised during the review process.

We look forward to receiving your revised manuscript.

Kind regards,

Mu-Hong Chen, M.D., Ph.D.

Academic Editor

PLOS One

Journal Requirements:

Reviewers' comments:

Reviewer's Responses to Questions

**Comments to the Author**

1. If the authors have adequately addressed your comments raised in a previous round of review and you feel that this manuscript is now acceptable for publication, you may indicate that here to bypass the “Comments to the Author” section, enter your conflict of interest statement in the “Confidential to Editor” section, and submit your "Accept" recommendation.

Reviewer #1: All comments have been addressed

Reviewer #2: All comments have been addressed

Reviewer #4: (No Response)

2. Is the manuscript technically sound, and do the data support the conclusions?

Reviewer #1: Yes

Reviewer #2: (No Response)

Reviewer #4: Partly

3. Has the statistical analysis been performed appropriately and rigorously? 

Reviewer #1: Yes

Reviewer #2: (No Response)

Reviewer #4: No

4. Have the authors made all data underlying the findings in their manuscript fully available?

Reviewer #1: (No Response)

Reviewer #2: (No Response)

Reviewer #4: Yes

5. Is the manuscript presented in an intelligible fashion and written in standard English?

Reviewer #1: (No Response)

Reviewer #2: (No Response)

Reviewer #4: Yes

6. Review Comments to the Author

Reviewer #1: The authors have comprehensively addressed all the comments and concerns raised in my previous review of the manuscript

Reviewer #2: (No Response)

Reviewer #4: I thank the authors for their thoughtful responses and acknowledge both the value and rarity of the dataset, as well as the practical and ethical constraints of conducting research in a forensic outpatient setting. Considering these issues, I accept the authors’ clarifications regarding the limitations of causal inference in the single-case design, the exploratory nature of the study and its implications for sample size, and the scope of the qualitative analyses. However, I remain concerned that the response to the issue of multiple statistical comparisons does not adequately address the core methodological concern raised in my original review.

While the authors argue that conventional corrections for multiple testing (e.g., Bonferroni) are not applicable within an idiographic Single-Case Experimental Design (SCED) framework, this argument does not sufficiently resolve the underlying issue. The concern is not the specific choice of correction method per se, but rather the extensive number of inferential statistical comparisons conducted across phases and outcome variables, and the absence of a clearly articulated strategy to manage, mitigate, or transparently acknowledge the resulting inflation of Type I error risk.

Simply stating that Bonferroni-type corrections are incompatible with SCED methodology is insufficient when numerous statistical tests are nevertheless reported and interpreted. At present, the manuscript does not clarify how multiplicity is handled analytically, whether alternative safeguards (e.g., emphasis on visual analysis, effect size patterns, consistency across phases, or other SCED-appropriate approaches) are employed, or how readers should interpret statistical significance in light of this multiplicity.

I therefore strongly encourage the authors to revisit and re-analyze the data using an explicit and methodologically coherent strategy to address the issue of multiple comparisons, or to substantially revise the analytical framework and interpretation to ensure that the risk of false-positive findings is appropriately controlled or transparently contextualized within the logic of SCED research.

7. PLOS authors have the option to publish the peer review history of their article (what does this mean? ). If published, this will include your full peer review and any attached files.

**Do you want your identity to be public for this peer review?** For information about this choice, including consent withdrawal, please see our Privacy Policy .

Reviewer #1: No

Reviewer #2: No

Reviewer #4: No

---

## [Author Response · Author response to Decision Letter 2]

12 Jan 2026

We thank the editor and the reviewers for taking the time to carefully re-evaluate our revised manuscript and our responses to the previous review round. We are pleased that the revisions and clarifications have addressed the reviewers’ earlier concerns. Below, we provide our responses to the remaining review comments. All changes have been incorporated into the revised manuscript, with page numbers indicated where relevant.

Review Comments:

Comment reviewer #1: We thank the reviewer for their careful re-evaluation of the revised manuscript and for this positive assessment.

Comment reviewer #4: We acknowledge that the number of statistical comparisons reported across phases and outcome variables increases the nominal probability of Type I error when p-values are considered in isolation. Within the logic of idiographic Single-Case Experimental Designs (SCEDs), however, these analyses do not constitute a single family of hypothesis tests aimed at population-level inference. As stated in foundational SCED methodology (e.g., Kratochwill et al., 2013; Krasny-Pacini & Evans, 2018), each comparison pertains to a specific participant, outcome, and phase contrast, and is intended to support descriptive and exploratory inference rather than confirmatory statistical testing.

To address multiplicity in a methodologically coherent manner, statistical significance testing is therefore not interpreted as a primary decision criterion. Instead, inference is based on the convergence of multiple, SCED-appropriate sources of evidence, including: (a) systematic visual inspection of level, trend, and variability across phases; (b) non-overlap–based effect size estimates (TAU-U), which quantify the degree and direction of change between phases while accounting for baseline trend; and (c) consistency of effect patterns across phase contrasts and participants.

Within the manuscript, p-values are reported for transparency but are interpreted descriptively and in conjunction with visual and effect size evidence, rather than dichotomously. Readers are explicitly cautioned in the discussion (page 32) against interpreting isolated statistically significant findings as confirmatory evidence. Instead, the strength of inference is derived from the replication logic of SCEDs and the convergence of visual and quantitative indicators.

---

## [Decision Letter · Decision Letter 2]

22 Jan 2026

‘I Gotta Feeling’: Exploring the Effects of a Smartphone app (Feelee) to Enhance Adolescents’ Emotion Regulation in Forensic Outpatient Settings. A Multiple Single-Case Experimental Design.

PONE-D-25-44103R2

Dear Dr. Merel Leijse,

We’re pleased to inform you that your manuscript has been judged scientifically suitable for publication and will be formally accepted for publication once it meets all outstanding technical requirements.

Kind regards,

Mu-Hong Chen, M.D., Ph.D.

Academic Editor

PLOS One

Additional Editor Comments (optional):

Reviewers' comments:

Reviewer's Responses to Questions

**Comments to the Author**

1. If the authors have adequately addressed your comments raised in a previous round of review and you feel that this manuscript is now acceptable for publication, you may indicate that here to bypass the “Comments to the Author” section, enter your conflict of interest statement in the “Confidential to Editor” section, and submit your "Accept" recommendation.

Reviewer #4: All comments have been addressed

2. Is the manuscript technically sound, and do the data support the conclusions?

Reviewer #4: Yes

3. Has the statistical analysis been performed appropriately and rigorously? 

Reviewer #4: Yes

4. Have the authors made all data underlying the findings in their manuscript fully available?

Reviewer #4: Yes

5. Is the manuscript presented in an intelligible fashion and written in standard English?

Reviewer #4: Yes

6. Review Comments to the Author

Reviewer #4: Thank you for the clarification. The explanation, grounded in established SCED methodology, adequately addresses my concern about multiple comparisons, and I have no further comments on this issue.

7. PLOS authors have the option to publish the peer review history of their article (what does this mean? ). If published, this will include your full peer review and any attached files.

**Do you want your identity to be public for this peer review?** For information about this choice, including consent withdrawal, please see our Privacy Policy .

Reviewer #4: No

---

## [Editor Report · Acceptance letter]

PONE-D-25-44103R2

PLOS One

Dear Dr. Leijse,

I'm pleased to inform you that your manuscript has been deemed suitable for publication in PLOS One. Congratulations! Your manuscript is now being handed over to our production team.

Kind regards,

on behalf of

Dr. Mu-Hong Chen

Academic Editor

PLOS One